# Perceived Interpersonal Distances: A Cross-Cultural Analysis of Young Taiwanese and Young Southeast Asians during the COVID-19 Pandemic

**DOI:** 10.3390/bs14010020

**Published:** 2023-12-27

**Authors:** Yi-Lang Chen, Yu-Chi Lee, Andi Rahman

**Affiliations:** 1Department of Industrial Engineering and Management, Ming Chi University of Technology, New Taipei 243303, Taiwan; m09218051@mail2.mcut.edu.tw; 2Department of Industrial Engineering and Management, National Taipei University of Technology, Taipei 106344, Taiwan; yclee@ntut.edu.tw; 3Department of Industrial Engineering, Andalas University, Padang 25175, Indonesia

**Keywords:** interpersonal distance, cross cultures, mask wearing, participant gender, target gender

## Abstract

Prior research on interpersonal distance (IPD) has predominantly concentrated on specific nationalities or population groups within their respective regions. There is a dearth of studies investigating IPD differences among individuals of distinct nationalities coexisting in the same geographical location. This study aimed to examine the variances in IPD between 100 young Taiwanese participants (comprising 50 males and 50 females) and 100 Southeast Asian individuals (including 50 males and 50 females). This study also considered factors affecting IPD, including target genders and mask-wearing conditions. The results of the four-way ANOVA indicate that target gender and mask-wearing conditions had a significant impact on IPD (*p* < 0.001). While there were no significant main effects for region and participant gender, there was a noteworthy interactive effect between these two variables on IPD. In general, Southeast Asian participants exhibited lower sensitivity to changes in IPD in response to the independent variables in comparison to their Taiwanese counterparts; in certain instances, their IPD did not notably increase when confronted with targets not wearing masks. While prior research typically indicated that women tend to maintain larger IPD than men, the current study observed this gender difference only among young Taiwanese participants. However, such a gender gap was absent among young individuals from Southeast Asia.

## 1. Introduction

While discussions on interpersonal distance (IPD) have a long history, dating back to the work of Hall in 1966 [1], recent years have seen a surge in interest and investigation, primarily due to the global COVID-19 pandemic. Governments worldwide have strongly recommended measures such as maintaining social distancing and wearing masks, especially in public spaces, as the primary strategies for epidemic prevention [2,3,4]. The impact of the COVID-19 pandemic on IPD has been extensively studied, taking into consideration factors such as participant gender [4,5,6], target gender [4,7,8], mask wearing [3,7,9,10], and vaccination [4]. Generally, men exhibited shorter IPD, and IPD is shorter when encountering fully vaccinated women who wear masks compared to their counterparts [4,5,6,7,8,9,10].

Prior to the COVID-19 pandemic, research on IPD primarily centered around interpersonal relationships. The theory of IPD, initially introduced by Hall [1], posits that the distance individuals maintain when interacting with others is not solely determined by their personal feelings towards the other person. Instead, it is also influenced by various factors, including certain dyad characteristics, such as gender or age, as well as the broader social environment in which the interaction occurs. Research has, in fact, confirmed that gender can play a role in influencing one’s preferred IPD [11,12,13]. Moreover, age appears to be a significant factor in predicting dyadic distances [12,13,14], with younger individuals generally preferring closer IPDs than their older counterparts. This observation raises the possibility that during times of epidemic, younger people may place themselves in situations closer to potential sources of infection.

Cultural norms are widely acknowledged as some of the most significant factors shaping one’s preferred IPD. As emphasized by Hall [1], what may be categorized as intimate in one culture may be perceived as personal or social in another. Hall emphasized the existence of specific customs governing spatial behavior. Sorokowska et al. [15] conducted a survey that encompassed 8943 participants from 42 different countries, and their findings indicated that individual characteristics, including age and gender, have a discernible impact on IPD preferences. Additionally, variations in these preferences were linked to the temperature in a given region. For instance, when interacting with a stranger, it was noted that people in countries with the shortest and longest IPDs were Argentina and Romania, respectively. Their average IPDs were approximately 78 cm and 135 cm, resulting in a notable difference of 1.75 times. During the COVID-19 pandemic, Gokmen et al. [16] delved into the data collected by Sorokowska et al. [15] and revealed that IPD preferences among countries, categorized by varying dimensions, significantly influenced the rate of COVID-19 spread in those nations. This underscores the profound impact of cultural factors on IPD and how appropriate IPD measures can effectively control the transmission of the COVID-19 virus.

Studies focusing on cultural differences typically involve comparisons of IPD among individuals from various cultural backgrounds or countries. For example, Remland et al. [17] highlighted distinctions in touch-related behavior between contact cultures in southern Europe and non-contact cultures in northern Europe. Li [18] examined the proxemic responses of Singaporean Chinese toward male targets from four different ethnicities, noting the shortest IPD when individuals of the same ethnicity were involved. Ozdemir [12] investigated the relationship between culture and IPD by observing behavior in four shopping malls in Turkey and the United States, revealing that male–female pairs interacted more closely than male–male and female–female pairs in all settings. Sicorello et al. [19] found that Japanese participants preferred larger overall IPD compared to their German counterparts, with female–female dyads exhibiting the smallest distances. Furthermore, Lee and Chen [6] investigated IPD between Chinese and Taiwanese individuals during the COVID-19 pandemic, revealing that Chinese participants perceived shorter IPD. Each of these studies serves a specific research purpose, such as validating cultural norms (e.g., contact and non-contact cultures as proposed by Hall [1]), comparing cultural differences, and exploring behavior in public spaces across different countries.

The outbreak of the COVID-19 pandemic has generally led to an increase in IPD, and various factors have been found to exert varying degrees of influence on IPD [2,3,7,20,21]. It is important to note that the impact of cultural differences on IPD carries different implications during the pandemic compared to the pre-pandemic era. Previously, the focus was primarily on comparing the differences in perceived IPD among individuals from diverse cultural backgrounds, such as varying countries. However, with globalization, particularly during the pandemic, people from different cultures have come together in shared spaces for work, study, and daily life. Differences in IPD among these various cultures can directly affect the development and effectiveness of measures for epidemic prevention.

In this study, we investigated the impact of participant gender, target gender, and mask wearing on the perception of IPD in a cohort of 100 young Taiwanese individuals and 100 young Southeast Asians. During the peak period of the COVID-19 pandemic, these 200 young students maintained close contact with their university campus on a daily basis. Consequently, exploring the shifts in IPD in relation to these variables within the context of two distinct ethnicities and their interactions can offer valuable insights into human social behavior during pandemic situations, such as the COVID-19 crisis. For Southeast Asian students who have left their home countries to pursue studies at universities in Taiwan, it is essential for them to exercise increased caution regarding the risks of the epidemic to ensure their own protection. Despite the fact that all 200 participants hail from the same Southeast Asian region, the diverse cultural influences within individual countries may still impact IPD. Consequently, our goal was to elucidate how various variables influence IPD between these two regional populations amid the COVID-19 pandemic. Building on the aforementioned considerations, we posit the following research hypotheses:

**Hypothesis 1** **(H1):**
*Southeast Asian participants exhibit a larger IPD than Taiwanese participants.*


**Hypothesis 2** **(H2):**
*Male participants maintain a larger IPD than female participants.*


**Hypothesis 3** **(H3):**
*Interacting with male targets results in a larger IPD than interacting with female targets.*


**Hypothesis 4** **(H4):**
*Interacting with unmasked targets leads to a larger IPD than interacting with masked targets.*


## 2. Methods

### 2.1. Participants

A total of 200 participants were enrolled in this study. Among them, 100 were Taiwanese, comprising 50 males and 50 females. The male participants had an average (standard deviation, SD) age of 22.1 (1.8) years and height of 171.2 (4.4) cm, while the female participants had an average (SD) age of 21.8 (1.9) years and height of 161.5 (4.1) cm. The remaining 100 participants included 50 males and 50 females from Southeast Asian countries. These Southeast Asian individuals were foreign students at Ming Chi University of Technology, Taiwan, with 40% originating from Indonesia, 29% from Vietnam, 21% from Thailand, and the rest hailing from other Southeast Asian nations. The male participants from this group had an average (SD) age of 23.4 (2.5) years and height of 169.3 (5.7) cm, whereas the female participants had an average (SD) age of 22.6 (1.8) years and height of 159.6 (4.3) cm. Data collection occurred between May and July 2022, during the peak of the COVID-19 pandemic in Taiwan. All participants had normal vision, reported no cognitive or mental health issues, were right-handed, and were not acquainted with the individuals they interacted with during the experiment. The study was granted approval by the Ethics Committee of Chang Gung Memorial Hospital, Taiwan. Participants were extensively briefed about the testing procedure and expressed their consent by signing a consent form before data collection began.

### 2.2. Experimental Setting

To mitigate the risk of human-to-human transmission during the COVID-19 pandemic, we utilized an online survey as a data collection method, a precautionary approach recommended by Calbi et al. [22]. This online survey was adapted from a paper-and-pencil test used in Hayduk’s study [23]. Online surveys have demonstrated their effectiveness in gathering IPD data and have become widely employed in both clinical and practical research settings [13]. The survey was conducted using a computer equipped with the Axure RP rapid prototyping tool (Axure RP 10 Software Solutions in San Diego, CA, USA).

During the assessment, participants were instructed to manipulate a virtual subject (avatar) using a cursor, guiding it towards a target, as depicted in Figure 1. Once the participants initiated the avatar’s movement, the directional arrow indicating the movement between the two avatars was concealed to ensure it did not influence their distance judgment. Essentially, no visual cues were provided to indicate the distance between the two avatars during the IPD determination, except for the perceptual changes caused by moving the avatar. Participants were tasked with visualizing and determining the IPD by positioning the avatar at a point that felt comfortable but had just started to become uncomfortable. This IPD definition is consistent with previous studies [5,6,23,24]. Subsequently, the distance between the two avatars was converted at a ratio of 1:7.2 to calculate the psychological IPD. The initial distance between the two avatars was initially set at 55.5 cm, representing an approximate initial distance of 4 m in the real world between the participant and the target [5,6]. To assess the measurement’s reliability, a pilot study was conducted, and the intraclass correlation coefficient between repetitions yielded a satisfactory reliability level of 0.85.

### 2.3. Targets

Two individuals, a 22-year-old male and a 22-year-old female, both with typical Taiwanese features, were selected as the targets for this study. This is because the study simulated an IPD examination on a university campus where the majority of students are Taiwanese. Both Taiwanese and Southeast Asian students can easily discern the distinctions in appearances between Taiwanese and non-Taiwanese individuals. The male had a height of 176 cm, while the female stood at 160 cm (Figure 2). These targets were dressed in everyday attire without any additional accessories. In order to create digital representations of these targets for the online survey, a digital camera (Sony HDR-XR260; Sony, Tokyo, Japan) was utilized to capture sagittal views of both the male and female targets in two distinct mask-wearing conditions. Throughout the image capture process, the targets were instructed to maintain a neutral expression. Subsequently, these photographs were integrated into the online survey. The digital representations of the male and female targets as displayed on the screen were proportionally scaled down to dimensions of 24.4 cm and 22.1 cm, respectively. The surgical masks used in the study were standard blue masks, devoid of any decorative elements, consistent with the type of face masks typically recommended during the COVID-19 pandemic.

### 2.4. Procedure and Design

Before commencing data collection, a detailed explanation of the testing procedure was provided to the participants by an experimenter. To aid participants in recalling their experiences during the COVID-19 pandemic, a 2 min video produced by Stanford Medicine was presented to introduce the pandemic. In addition, participants were presented with four images of the targets in 2 × 2 combinations, as depicted in Figure 2. These images were intended to assist participants in mentally simulating their feelings when facing the target under different scenarios, ensuring the quality of the IPD data. Each participant was required to complete three separate trials, and the average values from these trials were computed for subsequent analysis. To prevent participant fatigue, a minimum rest period of 3 min was provided between the trials. The trials were presented sequentially for IPD assessment and were randomly ordered.

To determine the IPD, participants utilized the computer mouse to adjust the position of their avatar until they felt it was close to the point of discomfort but still within a comfortable range, as shown in Figure 3. Participants had the flexibility to make minor adjustments to confirm their perceived distance. Once the participant had established their IPD, the computer automatically calculated and recorded the distance between the chins of the two avatars, following the method outlined in Chen and Rahman [4]. In total, 2400 data samples were collected, resulting from the combination of 2 testing groups, 100 participants, 2 target genders, 2 mask conditions, and 3 repetitions.

### 2.5. Statistical Analysis

The independent variables in the study were region, participant gender, target gender, and face mask wearing. The dependent variable under scrutiny was IPD, measured in centimeters. Data analysis was carried out using SPSS version 23.0 (IBM, Armonk, NY, USA), with a predefined significance level (α) of 0.05. A four-way analysis of variance (ANOVA) was conducted to assess the impact of the independent variables on IPD. Additionally, two separate three-way ANOVAs were executed for each of the regional participant groups. Post hoc comparisons were performed using independent-samples *t*-tests. Effect sizes were quantified using the η^2^ value for each effect, following Cohen’s guidelines [25]. Before conducting the statistical tests, the Kolmogorov–Smirnov test was employed to evaluate the adherence of numerical variables to a normal distribution, and Levene’s test was used to confirm the homogeneity of variances.

## 3. Results

Table 1 displays the outcomes of the four-way ANOVA conducted for the IPD measurements. The results revealed that target gender (H3) and mask wearing (H4) had a significant impact on IPD (*p* < 0.001), while region (H1) and participant gender (H2) did not exhibit any differences in IPD. Figure 4 represents the main effects of the four independent variables and the statistical paired comparisons. In general, participants reported a comparatively larger IPD when encountering male or unmasked targets. Notably, the interaction between region and participant gender demonstrated a significant effect (H1) and (H2), necessitating further cross-analysis to explore this interaction. It is noteworthy to mention that there was a statistically significant interaction effect between region and mask wearing (*p* < 0.05), albeit with a small effect size (η^2^ < 0.01). Under the unmasked condition, Southeast Asians tended to maintain a larger IPD than the Taiwanese, with a difference of approximately 5 cm. In summary, one of our study hypotheses was rejected (H1), while another was partially accepted (H2), and two were fully accepted (H3 and H4).

As presented in Table 2, the results of the three-way ANOVA indicate that the gender of Taiwanese participants significantly influenced IPD (*p* < 0.001), while the gender of Southeast Asian participants did not yield significant differences in IPD. Figure 5 provides further insight into the interaction between region and participant gender. Specifically, it seems that among Taiwanese participants, women reported a preference for a larger IPD compared to men (*p* < 0.001). However, among Southeast Asian participants, there was a non-significant trend in the opposite direction (*p* = 0.066). Figure 6 illustrates all paired comparisons for the four variables and their independent test results. Notably, Southeast Asian participants appeared to be less sensitive to IPD than their Taiwanese counterparts.

## 4. Discussion

It is widely recognized that different cultures or countries can influence IPD. During the peak of the epidemic, individuals from diverse cultural backgrounds interacted intensively in an environment, and variations in their perceived IPD could impact the overall development of the epidemic. This study aimed to explore differences in IPD among 100 young Taiwanese individuals and 100 young Southeast Asian individuals (foreign students residing in Taiwan). Although there was no distinction in overall IPD, different participant genders exhibited varying preferences for IPD. In summary, the study rejected H1, partially accepted H2, and fully accepted H3 and H4.

Specifically, Taiwanese young women maintained a larger IPD (*p* < 0.001), while Southeast Asian young women displayed a contrasting trend compared to men, although it did not reach statistical significance (*p* = 0.066). Zhou et al. [26] reported a connection between social interaction distance and perceptual judgments on social grouping, noting that female participants tended to maintain a greater distance in mixed-sex dyads due to feelings of insecurity and shyness [27]. However, in our study, this phenomenon was observed exclusively among Taiwanese participants. This suggests that, in comparison to young Taiwanese women, who tend to be more conservative in their approach to strangers, there was no significant difference between male and female students from Southeast Asia. Moreover, while both groups exhibited a preference for larger IPD when encountering male and masked targets, cross analyses revealed that only Southeast Asian female participants displayed a significantly larger IPD when facing unmasked male targets (Figure 6). In contrast, Taiwanese participants exhibited significant effects of mask wearing on IPD for both genders. Despite findings from a comprehensive cross-national study (n = 14,000) indicating a 54% global increase in preferred IPD during the COVID-19 pandemic across all types of relationships and countries [28], our study observed that the IPD was still influenced by various determining factors between different regional populations.

Table 1 reveals that neither region nor participant gender individually affects IPD, but there is an interaction between these two variables. While Sorokowska et al. [15] conducted a global comparative study that did not include Taiwan, Vietnam, Thailand, and other Southeast Asian countries, its analysis results suggested that Southeast Asian countries such as Indonesia, Malaysia, China, and Hong Kong all belong to relatively conservative regions, characterized by larger IPDs, with small differences among them. Therefore, our study finding of no significant difference in IPD between the two samples is in line with this inference. In the study results by Sorokowska et al. [15], the IPD data for Southeast Asians when encountering a stranger ranged from 110 cm for Indonesians to 115 cm for Hong Kong people. It is worth noting that in our study, the average IPDs across other variables (i.e., participant gender, target gender, and mask wearing) were approximately 171 cm and 174 cm for Taiwanese and Southeast Asians, respectively. This suggests that the difference of about 60 cm from the findings of Sorokowska et al. could be attributed to the impact of the COVID-19 pandemic, aligning closely with the 54% global increase in preferred IPD [28]. However, if the mask effect were deducted, the increase in IPD would be even more significant. Another factor influencing IPD was the disparity between actual measurement and online simulation. Kühne and Jeglinski-Mende [29] found that participants tended to overestimate the distance in pictures at an IPD of 150 cm compared to 50 and 90 cm. This discrepancy may be due to the wider distance between individuals not being perceived as dangerous. This limitation in virtual IPD measurement warrants attention.

Hall [1] classified cultures into two distinct categories: contact and non-contact cultures. Contact cultures prefer closer IPDs and engage in more physical touching, while people in non-contact cultures exhibit contrasting preferences and behaviors. In our study, both Taiwan and Southeast Asia can be categorized as non-contact cultures according to Hall’s classification [1]. While previous research typically compared the differences in IPD between Western and Eastern populations [12,30,31,32], our participant samples may be more consistent in their non-contact cultural traits. However, variations in IPD between the different variables were still observed.

In contrast to previous studies that often involved participants and targets of the same nationality, this study employed Chinese as targets. Consequently, when Southeast Asian participants encountered these targets, they were essentially facing “foreigners”. Li [18] conducted a study in which 173 Chinese Singaporean undergraduates rated the minimum IPD for perceived male intruders from four different ethnic groups in Singapore, namely Malay, Indian, Chinese, and Caucasian. The results of the rated distance scores revealed that Chinese–Chinese dyads exhibited the shortest IPD. This might lead one to expect that Southeast Asians in our study would determine their IPD to be larger. However, this did not seem to be the case. One possible explanation for this is that the foreign students in our study had been living and studying with Taiwanese students for a period of time, which could have reduced the sense of unfamiliarity between individuals of different races. Additionally, Lee and Chen [6] examined IPD between Chinese and Taiwanese individuals during the COVID-19 pandemic and found that Chinese participants perceived shorter IPD. The discrepancy in IPD observed in their study might be attributed to the fact that the tests were conducted in mainland China and Taiwan, respectively. Different environments and the varying stages of the epidemic’s development could potentially account for the distinct results. Another potential explanation for the findings could be derived from Pandey and Yu’s survey analysis [33] on the experiences of foreigners residing in Taiwan during the COVID-19 pandemic outbreak. Their study concluded that the experiences of foreign residents in Taiwan during this period were notably positive. Foreign residents expressed feeling comfortable, safe, and happy to stay and work in Taiwan, attributing these sentiments to the Taiwanese government’s successful policies in preventing community outbreaks.

While this study primarily focused on disparities in IPD preferences between two regional samples, the results indicated that target gender and the decision to wear a mask significantly influenced IPD in both groups, as illustrated in Table 2. Existing research concentrating on the impact of mask wearing on IPD during the COVID-19 pandemic has consistently shown a decrease in IPD when individuals are faced with a mask-wearing target [3,4,6,7,34], aligning with our findings. Cartaud et al. [11] reported a significant reduction in IPD when targets wore face masks, attributing this to a perception of increased trustworthiness compared to other conditions. Zhang et al. [35] used depth detection devices to analyze close contact behaviors in railway carriages and surrounding spaces, finding that when all passengers wore surgical masks, personal virus exposure through close contact could be reduced by approximately 52%. The presence of a mask led to a subjective perception of increased safety, resulting in a reasonable reduction in IPD.

Studies examining the impact of gender dyads on IPD have yielded diverse results. Yu et al. [5] observed that male dyads reported the greatest IPDs, while female dyads reported the shortest IPDs, consistent with findings in other studies [36,37]. In contrast, Hecht et al. [38] reported that IPDs in mixed-sex dyads were not significantly different from those in same-sex dyads. In our study, both Taiwanese and Southeast Asian participants exhibited shorter IPD when facing female targets. The interaction effects of participant gender and target gender were not significant (*p* = 0.091 for Taiwanese and *p* = 0.585 for Southeast Asians), as presented in Table 2. This suggests that there were no regional differences in IPD concerning the target gender variable.

This study has several limitations. Due to the pandemic, an online survey was utilized, and the IPD data collected may not perfectly align with data obtained in real-world settings. Additionally, the study exclusively employed blue surgical masks, leaving the effects of different types and colors of face masks on IPD perception unexplored. The Southeast Asian participants in this study encompassed foreign students in Taiwan from countries such as Indonesia, Vietnam, and Thailand, among others, in varying proportions. While we assumed minimal differences between them and disregarded them, future research could delve deeper into potential variations among these subgroups. It is also worth noting that, in the test, Taiwanese participants were slightly younger (average by 1 year) and taller (average by 2 cm) than Southeast Asian participants. While this study focused on distance-based measurements, these slight differences, especially in height, may impact the results. Moreover, because comparative investigations in IPD between specific regional populations are relatively scarce, this study cannot directly compare IPD results with those from previous studies. Given that IPD has changed as the epidemic has evolved [20,28], the comparison of absolute IPD values may lack significant meaning. Finally, the challenge with exploratory research like this is that, given the large number of findings tested (in a 2 × 2 × 2 × 2 ANOVA with four potential main effects and eleven potential interactions), there is a high risk of false positives. Ideally, these findings should be replicated to confirm their existence, although it may be challenging due to the evolving nature of IPD during the epidemic.

## 5. Conclusions

Past studies on IPD have often focused on specific nationalities or populations within their respective regions. However, studies that examine differences in IPD between individuals from different nationalities sharing the same location hold valuable insights for epidemic control strategies. This study sought to understand the variations in IPD between young Taiwanese and Southeast Asian individuals concerning different target genders and various mask conditions. The results show significant effects of target gender and mask wearing on IPD in both regional populations, indicating broader implications beyond the specific regions studied. Specifically, a female target led to a shorter IPD compared to a male target. Importantly, when faced with a mask-wearing target, individuals make shorter IPD decisions due to feeling relatively safe. These two convincing findings carry significant implications for managing future similar epidemics and comprehending the essence of IPD.

In our results, although there were no significant main effects for region and participant gender, an interaction effect was observed. In general, Southeast Asian participants appeared to be less sensitive to IPD changes in response to the independent variables compared to their Taiwanese counterparts. Furthermore, in some instances, their IPD did not significantly increase when facing targets not wearing masks. Additionally, previous research typically showed that women maintained larger IPD than men. Similarly, young Taiwanese participants in this study exhibited this gender difference, but such a gender gap was not observed among young individuals from Southeast Asia. Given the limited number of relevant regional comparative studies on IPD, it is crucial to emphasize how future research can validate the current findings. This can be achieved through a large-N pre-registered study with specific hypotheses, ensuring a more robust confirmation of the present results.

## Figures and Tables

**Figure 1 behavsci-14-00020-f001:**
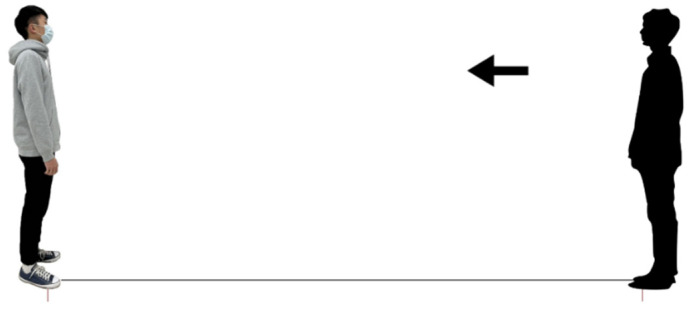
Screenshot of the online survey showing the participant approaching a male target wearing a surgical mask (Note: the arrow indicating participant movement direction [right avatar]).

**Figure 2 behavsci-14-00020-f002:**
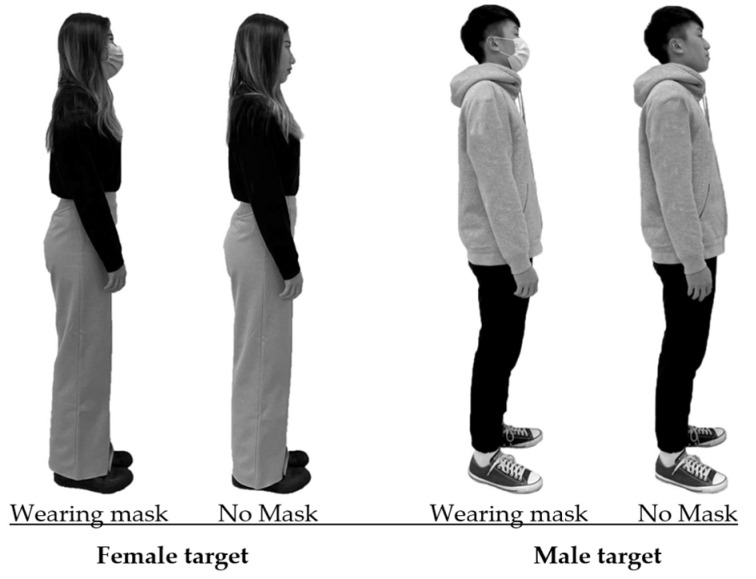
Images of the targets in different testing combinations (2 genders × 2 mask conditions).

**Figure 3 behavsci-14-00020-f003:**
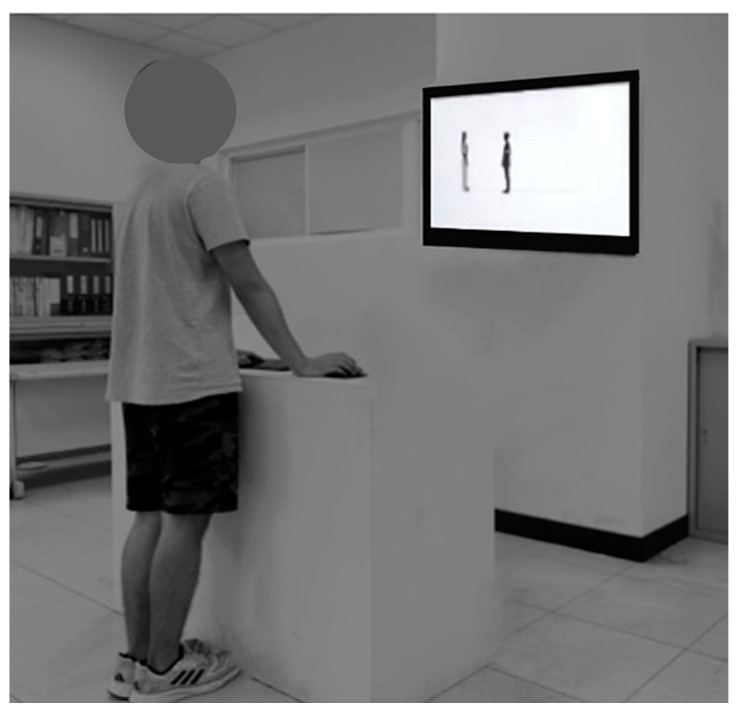
Schematic representation of the test layout and the testing situation.

**Figure 4 behavsci-14-00020-f004:**
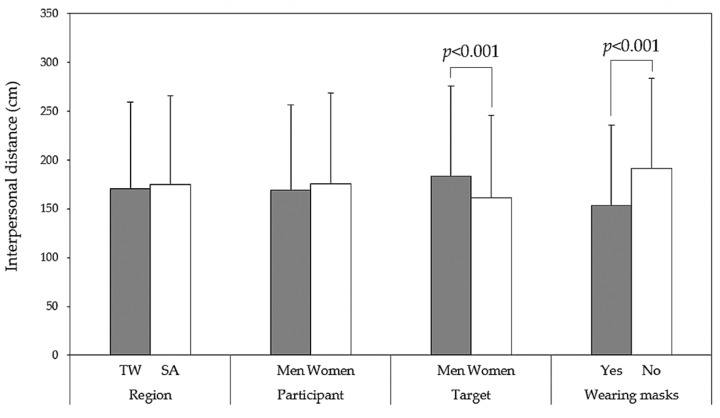
Main effect comparisons of four independent variables via independent *t*-tests (TW: Taiwanese; SA: Southeast Asians).

**Figure 5 behavsci-14-00020-f005:**
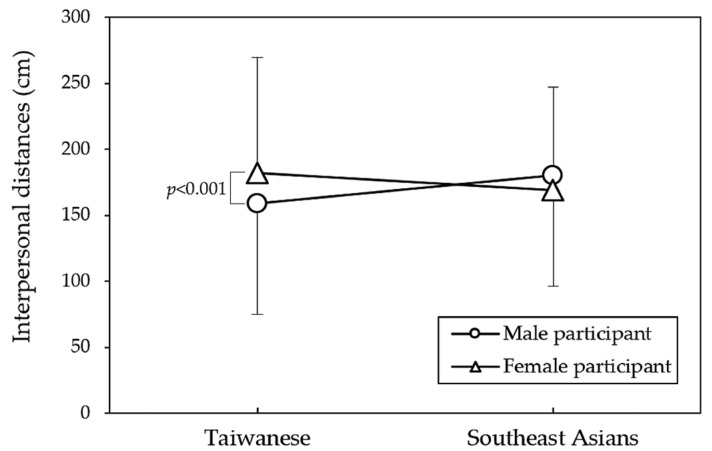
Interpersonal distance comparisons among participant genders in regional groups through independent *t*-tests.

**Figure 6 behavsci-14-00020-f006:**
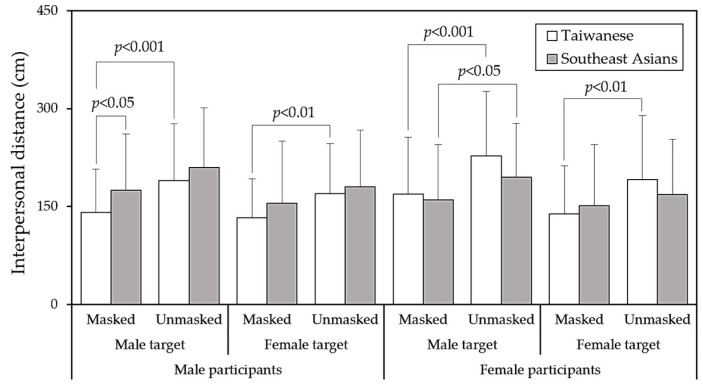
Pairwise comparisons in regional groups using independent *t*-tests for various test combinations.

**Table 1 behavsci-14-00020-t001:** Four-way ANOVA results of interpersonal distance for both regional participant groups.

Sources	SS	df	MS	F	*p*	η^2^
Region (R)	7604	1	7604	1.06	0.304	0.001
Participant gender (PS)	13,850	1	13,850	1.93	0.165	0.001
Target gender (TS)	204,896	1	204,896	28.53	<0.001	0.018
Mask (M)	595,231	1	595,231	82.87	<0.001	0.050
R × PS	118,629	1	118,629	16.52	<0.001	0.010
R × TS	636	1	636	0.09	0.766	<0.001
R × M	44,569	1	44,569	6.21	<0.05	0.004
PS × TS	4194	1	4194	0.58	0.445	<0.001
PS × M	13,396	1	13,396	1.87	0.172	0.001
TS × M	13,396	1	13,396	1.87	0.172	0.001
R × PS × TS	17,438	1	17,438	2.43	0.119	0.002
R × PS × M	6480	1	6480	0.90	0.342	0.001
R × TS × M	709	1	709	0.10	0.753	<0.001
PS × TS × M	7	1	7	<0.01	0.974	<0.001
R × PS × TS × M	1378	1	1378	0.19	0.661	<0.001

**Table 2 behavsci-14-00020-t002:** Three-way ANOVA results of interpersonal distance for each regional participant group.

Sources	Taiwanese (n = 100)			Southeast Asians (n = 100)		
	F	*p*	η^2^	F	*p*	η^2^
Participant gender (PG)	15.77	<0.001	0.020	3.39	0.066	0.004
Target gender (TG)	16.86	<0.001	0.021	12.02	<0.01	0.015
Mask (M)	71.30	<0.001	0.083	20.67	<0.001	0.026
PG × TG	2.86	0.091	0.004	0.30	0.585	<0.001
PG × M	1.14	0.286	0.001	0.09	0.765	<0.001
TG × M	0.59	0.444	0.001	1.33	0.248	0.002
PG × TG × M	0.09	0.768	<0.001	0.10	0.747	<0.001

## Data Availability

The data are available upon reasonable request to the corresponding author.

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
