# Peer review of "Perceived Interpersonal Distances: A Cross-Cultural Analysis of Young Taiwanese and Young Southeast Asians during the COVID-19 Pandemic"

_behavsci, 2023, doi:10.3390/bs14010020_

Round 1
Reviewer 1 Report
Comments and Suggestions for Authors
I enjoyed reviewing your study. I think it offers a compelling and thoughtful contribution to the literature. The only suggestion I have is to elaborate a bit more on interpersonal distance, looking for more recent citations that have examined proxemic zones.
The design of the experiment is well organized and easy to follow. The experiment itself is original and logically sound.
Reviewer 2 Report
Comments and Suggestions for Authors
Review of Manuscript #behavsci-2722156 submitted to Behavioral Sciences:
Perceived Interpersonal Distances: A Cross-Cultural Analysis of Young Taiwanese and Young Southeast Asians during the COVID-19 Pandemic
December 1st, 2023
The purpose of this submission was to study perceptions of appropriate interpersonal distance (IPD) in a simulated online setting among 200 Taiwanese and Southeast Asian male and female participants, utilizing online male and female targets who either were or were not wearing a mask. Participants preferred a substantially greater IPD for male (vs. female) targets, as well as for targets not wearing a mask (vs. targets wearing a mask). There was also an interaction between participants’ nationality and gender, such that Taiwanese females maintained a larger IPD than males, whereas for the Southeast Asian participants the gender difference trended non-significantly (p = .066) in the opposite direction.
The strength of this research is that it examines the influence of two independent variables (target gender and mask-wearing) as well as two participant variables (two Asian nationalities and participants’ gender) on IPD. It was particularly helpful to examine the influence of wearing masks on IPD during the COVID-19 pandemic.
The greatest weakness of this study is the fact that the hypotheses were apparently not specified ahead of time, and thus the findings are somewhat exploratory. There is no mention of preregistered hypotheses. In fact, the only stated hypothesis was that “the influence of different culture within distinct countries might still affect IPD,” but the direction of this influence was not specified. The problem with exploratory research and a lack of a theoretical basis for the findings is that, given the large number of findings tested (in a 2 x 2 x 2 x 2 ANOVA, this includes 4 potential main effects and 11 potential interactions), there is a high danger of false positives. This issue should definitely be addressed in the discussion as a limitation of the current research; ideally, these findings should be replicated in order to confirm their existence.
Due to the above issue, I find the two very large main effects (target gender and mask-wearing) to be the most convincing findings, since such huge effects would be less likely to occur due to chance. However, the Discussion section of this manuscript barely mentions these two large effects. I suggest focusing more on these two main effects and the reasons for them, as well as how they fit in with other IPD research.
Another important weakness to the current study is the fact that real-world preferred IPD was not assessed in this study. Since the study took place during the COVID-19 pandemic, it is understandable that the authors wished to conduct research online. However, when it comes to preferred IPD, it is possible that what people claim they would do is different than what they actually do in real life. Nevertheless, the authors deserve credit for acknowledging this as a limitation. Also, it is interesting to explore how people’s IPD “claims” (which is essentially what the current study measures) may differ from their actual IPD behavior. For that reason, again, it is important to discuss how the findings from this study compare to the findings on real-world IPD among east Asian populations.
I have just a few other smaller issues to mention, all of which should be relatively simple for the authors to address:
- The Taiwanese participants were slightly younger and taller than the Southeast Asian participants. While this may not have mattered, it is technically a confound, and it is theoretically possible that this could account for any differences in preferred IPD between the two nationalities studied. This should be briefly mentioned in the Discussion section as a limitation.
- In the first paragraph of the Results section (end of p. 5), I believe the finding regarding masked targets was misstated. The manuscript states that “participants reported a comparatively larger IPD when encountering male or masked targets.” But I believe participants actually had a much larger preferred IPD with unmasked (vs. masked) targets, correct? This finding has important implications for COVID-related behavior, so it should be clear (and as mentioned above, it should receive more focus in the discussion).
- At the very end of the first paragraph of the Results section (last sentence on p. 5), the authors mention a statistically significant interaction, but say that its impact can be considered negligible due to a small effect size. Since the effect is statistically significant, I still think a sentence should be added to this paragraph stating the direction of this region X mask-wearing interaction.
- It appears that instead of doing a post-hoc analysis on the region x participant gender interaction, the authors conducted two separate three-way ANOVAs (one for each region). This seems like an unusual way to examine the regional interaction in the four-way ANOVA, but I suppose it does lead to the same essential information that a post-hoc analysis would.
- Toward the end of p. 6, the authors state that, “Taiwanese females tended to maintain a larger IPD than males (p < 0.01), whereas no difference was observed between genders among the Southeast Asian participants.” Later, in the Discussion section (first new paragraph on p. 8), the authors state that “Taiwanese young women maintained a larger IPD (p < 0.01), whereas Southeast Asian young women displayed a contrasting trend, although it did not reach statistical significance (p = 0.463).” There are really two issues here. First, is the p-value of 0.463 correct? According to Table 2, shouldn’t the p-value be 0.066? This is certainly more consistent with the idea of a “trend” that “did not reach statistical significance.” Second, there is an inconsistency in the way this result is described on p. 6 vs. p. 8. I think that the description on p. 8 (after the p-value is corrected) is actually more accurate. It appears that for Taiwanese participants, women report preferring a larger IPD than men; but for Southeast Asian participants, there is a non-significant trend in the opposite direction (men preferring a larger IPD than women).
- At the end of the first new paragraph on p. 8, the authors state that “Southeast Asian participants generally had no significant impact on their perceived IPD whether the target wore a mask or not.” Are you sure this is correct? Table 2 seems to indicate that there is a large main effect of whether or not targets wore masks for both regions, though the effect is larger for Taiwanese participants.
- How obvious is the ethnicity of the virtual targets to Taiwanese and Southeast Asian participants? The authors mentioned that the targets had “typical Taiwanese features” (p. 4), and subsequently this fact is used to explain some of the regional effects (p. 8). I ask because I suspect most Westerners would not be able to distinguish the prototypical appearances of Taiwanese vs. Southeast Asian people. But as long as this feature is obvious to the participants in this study, it is reasonable to make assumptions about the effect of the targets’ features on the participants (p. 8).
In sum, this research adds to our knowledge regarding the influence of nationality, gender, and mask-wearing on participants’ claimed preferred IPD during the COVID-19 pandemic. The exploratory nature of this study (and lack of specific hypotheses), as well as the fact that the study examines claims about IPD rather than actual preferred IPD, are serious limitations. A stronger focus on the two very large main effects (which are unlikely to be false positives) is warranted. The authors may wish to conclude by focusing on exactly how future research can help confirm the present findings with a large-N preregistered study with specific hypotheses.
Reviewer 3 Report
Comments and Suggestions for Authors
Congratulations on a very interesting piece of research. As a limitiation of your study, you mention that your results are not compared to those of previous studies. I think your contribution will be far more interesting if you could include some comparison.
Round 2
Reviewer 2 Report
Comments and Suggestions for Authors
Second review of Manuscript #behavsci-2722156 submitted to Behavioral Sciences:
Perceived Interpersonal Distances: A Cross-Cultural Analysis of Young Taiwanese and Young Southeast Asians during the COVID-19 Pandemic
December 21st, 2023
This re-submission of a study on perceptions of appropriate interpersonal distance (IPD) in a simulated online setting shows substantial improvement over the first draft. The authors nicely addressed every point made in my initial feedback, and their response letter clearly described exactly where each improvement was made in the new draft. (I was not able to view feedback by other reviewers, if there was any.)
I have only a few small further suggestions:
First, the authors clear described their four hypotheses (labeled H1, H2, H3 and H4) immediately before the Methods section. My only additional recommendation regarding this point is to match each hypothesis with the corresponding statistical test(s) in the Results section, to make clear which analysis is testing which of the four hypotheses.
Second, in lines 277-285, the authors included a discussion of past research on preferred IPD. However, I believe the authors accidentally wrote the same sentence twice, with only a few words changed. One of the two sentences in this section should be omitted.
Third, in the second-to-last sentence of the Abstract (line 25), I recommend adding the word “only” immediately before “among young Taiwanese participants.”
Finally, I mentioned in my review that lack of preregistered hypotheses is a weakness to this study; however, I find the two very large main effects (target gender and mask-wearing) to be the most convincing findings, since such huge effects would be unlikely to occur due to chance. I believe the “Conclusions” paragraph (lines 379 to 393) should focus more on these two findings. Also, at the end of my review, I suggested that, “The authors may wish to conclude by focusing on exactly how future research can help confirm the present findings with a large-N preregistered study with specific hypotheses.” I still believe this point should be mentioned at some point in the final “Conclusions” paragraph.
If these changes are made, I support the publication of this article. This research adds to our knowledge regarding the influence of nationality, gender, and mask-wearing on participants’ claimed preferred IPD during the COVID-19 pandemic.
